# Multi-Modal Stacking Ensemble for the Diagnosis of Cardiovascular Diseases

**DOI:** 10.3390/jpm13020373

**Published:** 2023-02-20

**Authors:** Taeyoung Yoon, Daesung Kang

**Affiliations:** Department of Healthcare Information Technology, Inje University, 197, Inje-ro, Gimhae-si 50834, Republic of Korea

**Keywords:** deep learning, stacking ensemble, machine learning, cardiovascular diseases

## Abstract

Background: Cardiovascular diseases (CVDs) are a leading cause of death worldwide. Deep learning methods have been widely used in the field of medical image analysis and have shown promising results in the diagnosis of CVDs. Methods: Experiments were performed on 12-lead electrocardiogram (ECG) databases collected by Chapman University and Shaoxing People’s Hospital. The ECG signal of each lead was converted into a scalogram image and an ECG grayscale image and used to fine-tune the pretrained ResNet-50 model of each lead. The ResNet-50 model was used as a base learner for the stacking ensemble method. Logistic regression, support vector machine, random forest, and XGBoost were used as a meta learner by combining the predictions of the base learner. The study introduced a method called multi-modal stacking ensemble, which involves training a meta learner through a stacking ensemble that combines predictions from two modalities: scalogram images and ECG grayscale images. Results: The multi-modal stacking ensemble with a combination of ResNet-50 and logistic regression achieved an AUC of 0.995, an accuracy of 93.97%, a sensitivity of 0.940, a precision of 0.937, and an F1-score of 0.936, which are higher than those of LSTM, BiLSTM, individual base learners, simple averaging ensemble, and single-modal stacking ensemble methods. Conclusion: The proposed multi-modal stacking ensemble approach showed effectiveness for diagnosing CVDs.

## 1. Introduction

Cardiovascular diseases (CVDs) are a global public health problem and result from a variety of causes. Since CVDs are a disease of multifactorial origin, it is not easy to accurately and timely diagnose the disease [1]. Early and accurate diagnosis and treatment of CVDs can significantly reduce the risk of morbidity and mortality, making rapid and accurate CVDs prediction a crucial task in healthcare. Cardiologists use various tools to diagnose cardiovascular diseases, and one commonly used tool is the electrocardiogram (ECG). It enables quick detection of abnormal heart rhythms and potential heart disease signs without any intervention [2,3]. In particular, the most frequently used complementary exam for cardiac evaluation is a standard short-duration 12-lead ECG (S12L-ECG) since it can provide a comprehensive evaluation of the heart’s electrical activity. Therefore, the S12L-ECG system is used in various medical environments, ranging from primary care centers to intensive care units [4,5].

However, the ECG signal is complex and can be affected by various factors, such as noises and motion artifacts [6]. This makes it challenging to accurately diagnose CVDs. One way to overcome this limitation is to apply deep learning methods. Deep learning methods have been used to improve the accuracy of CVDs diagnosis by automatically learning features from the ECG signal that are relevant to the CVDs. When it comes to deep learning techniques utilized in detecting CVDs, recurrent neural networks (RNN), long short-term memory (LSTM), and gated recurrent units (GRU) have been extensively employed [7,8,9]. Faust et al. used a bidirectional LSTM (BiLSTM) to identify atrial fibrillation beats in heart rate signals, while Gao et al. proposed an LSTM that incorporated focal loss to address the imbalance of ECG beats [7,8].

Convolutional neural networks (CNN) are also widely used for diagnosing CVDs [10,11,12,13,14,15]. The one-dimensional CNN (1D CNN) model exploits the one-dimensional (1D) structure of the signal, so it can be used on these ECG data without transformation. The 1D CNN models, which are popular representation learning methods of 1D signals, can learn distinguishing hierarchical features when applying 1D convolution to 1D signals. The 1D CNN models hierarchically learn primitive features from the lower layers and complex features through consecutive higher layers [11]. Yildirim et al. constructed a 1D CNN and LSTM combination model to detect four and seven rhythm classes [11]. Mousavi et al. proposed a deep learning architecture that comprises the CNN layers, attention mechanism, and LSTM units to mitigate the occurrence of false alarms for arrhythmia detection in intensive care units [12].

Recently, there have been numerous studies conducted to detect CVDs using two-dimensional CNN (2D CNN) with ECG signals [14,15,16,17,18]. In order to apply a 1D ECG signal to a 2D CNN, the 1D signal needs to be transformed into a two-dimensional (2D) image. Jun et al. obtained 2D ECG images from 1D ECG signals by plotting each ECG beat as a grayscale image to classify eight rhythms [16]. In this study, we refer to the transformed 2D image as an ECG grayscale image. As another method, we can convert the 1D ECG signal into a spectrogram through a short-time Fourier transform (STFT) or a scalogram using wavelet transform. Yildirim et al. fine-tuned 2D CNN models (AlexNet, VGGNet, ResNet, and DenseNet) with spectrogram images to identify diabetes mellitus and Yoon et al. applied a pretrained ResNet-50 model to the ECG scalograms to classify four rhythms [14,15]. We refer to the converted scalograms as scalogram images. As another method, Zhai et al. used a 2D CNN architecture with a dual beat coupling matrix to identify supraventricular ectopic beats and ventricular ectopic beats [17]. 2D CNN has the advantage of utilizing pretrained models that were trained with a large number of images, such as the ImageNet database. In addition, there are several established 2D CNN architectures that have demonstrated good performance, so there is no need to design a new 2D CNN architecture by modifying layers and filters. To take advantage of the 2D CNN mentioned above, we aim to diagnose CVDs by fine-tuning a pretrained ResNet-50 model with scalogram images and ECG gray-scale images.

However, a single CNN model may not be sufficient to accurately predict CVDs, as it may suffer from high bias or high variance [19]. One way to address this problem is through ensemble methods, which combine the predictions of multiple single CNN models. Ensemble methods can lead to an improvement in performance by combining the strengths of multiple models and reducing the influence of their individual weaknesses. Ensemble models can also be used to reduce overfitting and improve generalization [20]. In this study, we aim to reduce the weakness of 12 individual ResNet-50 models for 12 ECG leads and enhance the strengths of those models using a simple averaging ensemble and stacking ensemble with two kinds of input modalities: scalogram image and ECG grayscale image. The two types of input images exhibit different characteristics. To obtain the characteristics of each image, we propose a multi-modal stacking ensemble that can utilize information obtained from different input modalities.

The major contributions of the study are outlined in the following manner: (1) for each lead, the performances of ResNet-50 based on scalogram images was compared to the performance of ResNet-50 based on ECG grayscale images; (2) we demonstrated that the diagnostic performance of the single-modal stacked ensemble was superior to that of the 12 individual base learners and single-modal simple averaging ensemble for both scalogram images and ECG grayscale images; (3) we proposed a multi-modal stacking ensemble that combines base learner predictions obtained from scalogram images and ECG grayscale images and then fed them as inputs to a meta learner; (4) the proposed multi-modal stacking ensemble demonstrated superior performance compared to LSTM, BiLSTM, 12 individual base learners, simple averaging ensemble, and the single-modal stacking ensemble.

## 2. Materials and Methods

### 2.1. Dataset and Preprocessing

The dataset used in this research was a 12-lead ECG database that was collected by Chapman University and Shaoxing People’s Hospital in China [6]. The 12-lead ECG database, which was recorded at a sampling frequency of 500 Hz, consisted of 10,646 patients (including 5956 males) and each recording lasted for 10 s. The ECG database contained 11 different heart rhythms labelled by professional physicians. Since raw ECG signals contain unwanted noise, the following three preprocessing steps were sequentially applied: Butterworth low-pass filter (LPF), local polynomial regression smoother (LOESS) curve fitting, and non-local means (NLM) technique [21,22,23]. The Butterworth LPF was used to remove signals with frequencies above the typical frequency range of a normal ECG (0.5 Hz to 50 Hz). To eliminate the baseline wandering effect that can be caused by respiration, the LOESS curve fitting method was used. The NLM technique was employed to reduce residual noises. Of the ECG data, 58 ECG recordings were excluded from the study since they either only had zeros or some of their channel values were incomplete. Among the remaining 10,588 data, the number of ECG samples with atrial tachycardia (AT), atrioventricular node reentry tachycardia (AVNRT), atrioventricular reentry tachycardia (AVRT), and sinus atrial-to-atrial wander rhythm (SAAWR) categories was only 121, 16, 8 and 7, respectively. The number of samples belonging to the four categories mentioned above was extremely small and hence excluded from this study. Finally, a sum of 10,436 ECG recordings belonging to 7 ECG rhythms were used in this study. Table 1 provides a comprehensive description of 7 distinct ECG rhythms along with the corresponding number of subjects.

### 2.2. Data Transformation

To utilize the 2D CNN model, it is necessary to transform the 1D ECG signal into a 2D image. Among the various methods of converting to a 2D image, we adopted a method of converting to a scalogram and a method of plotting a 1D ECG signal as it is in two dimensions. In this study, we refer to the former image as a scalogram image and the latter image as an ECG grayscale image. An ECG scalogram image is a visual representation of the time-frequency composition of the ECG signal that can reveal important information about the frequency characteristics of the ECG over time. Scalogram images were generated by applying the continuous wavelet transform (CWT) to the ECG recordings. An analytic Morse wavelet with a symmetry parameter of 3 (γ = 3) and a time-bandwidth product of 60 ($P^{2}=60$) was used to obtain the CWT. The Morse wavelet is perfectly symmetric in the frequency domain and has zero skewness when γ equals 3. The CWT was calculated using 10 voices per octave, a 500 Hz sampling frequency, and a signal length of 5000. The minimum and maximum scales were determined automatically based on the wavelet’s energy spread in time and frequency [24]. In this study, we used the *cwt.m* function provided by Wavelet Toolbox in Matlab 2020a (https://www.mathworks.com/help/wavelet/ref/cwt.html, accessed on 13 February 2023). The converted scalogram images were saved as 300 × 300 pixel RGB images. For ECG grayscale images, 1D ECG recordings were plotted as grayscale images with a white ECG signal against a black background. The ECG grayscale images were saved as 300 × 300 pixels. Examples of scalogram images and ECG grayscale images for the 7 groups (AFIB, AF, ST, SVT, SB, SR, and SI) are shown in Figure 1.

### 2.3. Ensemble Methods

Ensemble methods are a group of techniques that combine the predictions of multiple models to improve performance. There are many ensemble methods, but this study adopts simple averaging ensemble and stacking ensemble. Simple averaging ensemble obtains the output by averaging the predictions of individual learners directly. Owing to its simplicity and effectiveness, the method is popular in many real applications. The stacking ensemble consists of multiple base learners and a meta-learner. In stacking ensemble, each base learner trains with the original training dataset and then generates new datasets for training a meta learner, where the outputs of the base learner are regarded as input features of the meta learner. The stacking ensemble is powerful because it can combine the strengths of different models to produce a more accurate prediction [20].

Since we propose a multi-modal stacking ensemble method for diagnosing CVDs, we focus on a stacking ensemble. In this study, we use two types of image modalities: scalogram images and ECG grayscale images. The single-modal stacking ensemble refers to the stacking ensemble that utilizes only one image modality, whereas the multi-modal stacking ensemble refers to the stacking ensemble that incorporates two image modalities. We first explain the single modal stacking ensemble, and, to ensure clear understanding, we specifically describe the scenario where the input is a scalogram image. As shown in Figure 2, scalogram images are fed to a pretrained ResNet-50 model to be fine-tuned for each lead. Since we have 12 leads, 12 ResNet-50 base learners are fine-tuned with scalogram images. We can then obtain 12 predictions from 12 individual base learners. Each base learner’s prediction is a 7-dimensional probabilities vector. Considering 12 leads, we can obtain 12 predictions that consist of 7-dimensional probability vectors. Simple averaging ensemble averages the predictions of 12 single-lead ResNet-50 models that were independently trained. On the other hand, the stacking ensemble combines the predictions of the 12 base learners. That is, the 7-dimensional output probability vector from each lead is concatenated to make an 84-dimensional vector. Then the 84-dimensional vector is fed into a meta learner that outputs prediction values for the 7 ECG rhythms. As the meta learner, logistic regression, support vector machines (SVM), random forest, and XGBoost were employed in this study [25,26,27,28]. The single-modal stacking ensemble architecture for ECG grayscale images is the same as described above, except that the input image is an ECG grayscale image instead of a scalogram image.

Single-modal stacking ensemble considers only one input modality, whereas multi-modal stacking ensemble methods take multiple input modalities into account. A detailed description of the multi-modal stacking ensemble is depicted in Figure 3. In this study, scalogram images and ECG grayscale images are used as two input modalities. In the proposed multi-modal stacking ensemble, we combine an 84-dimensional vector obtained from 12 individual base learners using scalogram images and another 84-dimensional vector attained from 12 individual base learners using ECG grayscale images. Combining the vectors obtained from the two modalities results in a 168-dimensional vector. The concatenated 168-dimensional vector contains the characteristics of a scalogram image and an ECG grayscale image. The 168-dimensional vector becomes the new input vector for the meta learner. Similar to the single-modal stacking ensemble, the multi-modal stacking ensemble employed logistic regression, SVM, random forest, and XGBoost as meta learners in this study.

### 2.4. LSTM

LSTM is a type of recurrent neural network and a powerful method for the diagnosis of CVDs. By training an LSTM model on labeled ECG recordings, the model can learn to detect patterns and features that are indicative of CVDs. Due to the sequential nature of ECG recordings, LSTM is well-suited for this task as it can capture long-term and temporal dependencies between individual ECG recordings. In this study, LSTM was applied to the same ECG dataset to demonstrate the effectiveness of the proposed multi-modal stacking ensemble method. In experiment settings, LSTM has numerous hyperparameters; however, this study chose to set the batch size, hidden size, dropout, and number of epochs to fixed values of 128, 128, 0.2, and 100, respectively. The Adam optimizer was used with $\beta_{1}$ set to 0.9 and $\beta_{2}$ set to 0.999 to optimize the LSTM model. To determine the learning rate and number of layers, a grid search was performed where the learning rates were evaluated over the range of (1e-3, 1e-4, 5e-5, 1e-5), and the number of layers was tested within the range of (2, 3, 4). The best hyperparameter was chosen by selecting the one with the highest accuracy on the validation dataset. To prevent the vanishing gradient problem, the ECG signal sampled at 500 Hz was downsampled to 250 Hz. LSTM was trained for all 12 leads at the same time since a 12-lead ECG signal can be represented as a sequence of a 12-dimensional vector with a length of *T* time sample. On the other hand, ResNet-50 was trained individually for each lead. BiLSTM can be seen as a variation of LSTM. Unlike LSTM, BiLSTM can analyze input sequences both forward and backward, which gives it the ability to comprehend information from past and future time-steps and identify complex inter-dependencies in the data. BiLSTM was also experimented under the same conditions.

### 2.5. ResNet-50 Model and Machine Learning Algorithms

ResNet is a deep neural network architecture introduced in 2015. It was developed to address the issue of vanishing gradients that arises in deep networks. This problem is resolved by adding skip connections between the layers. The skip connection is a type of feedforward network that involves a shortcut connection. It adds new inputs to the network and yields new outputs, enabling the network to learn the residual mapping instead of the original mapping. ResNet has achieved state-of-the-art accuracy in a variety of computer vision tasks and became one of the most popular architectures for image classification and computer vision tasks [29]. For this reason, we used a pretrained ResNet-50 model as a base learner. To fine-tune the ResNet-50 model, we utilized the Adam optimizer with $\beta_{1}=0.9$ and $\beta_{2}=0.999$. The experiments were conducted with three initial learning rates (1e-4, 5e-5, 1e-5) of the Adam optimizer. Of the three learning rates, 5e-5 was adopted as the most accurate in the validation set among individual base learners. We fixed the mini-batch size at 32 and the number of epochs at 30. The ResNet-50 model was developed with a PyTorch framework [30]. The computer specifications used in the experiments are as follows: Intel Core i7-9700K 3.60GHz CPU, 64GB memory, and a 12GB NVIDIA GeForce GTX 2080 Ti graphics card. In this study, we considered four machine learning classifiers as a meta learner of the stacking ensemble: logistic regression, SVM, random forest, and XGBoost. We employed Scikit-learn library (https://scikit-learn.org/stable/index.html, accessed on 30 January 2023) to implement logistic regression, SVM, and random forest classifiers, while XGBoost was implemented using XGBoost Python Package (https://xgboost.ai/, accessed on 30 January 2023). Optimal hyperparameters for the meta learner were chosen by performing a thorough grid search and evaluating the accuracy of the validation set. The details of the hyperparameters which were tuned using the grid search are described in Table 2. The code for training and evaluating the proposed multi-modal stacking ensemble model is available at: https://github.com/xodud5654/MMSE (accessed on 17 February 2023).

## 3. Results

We evaluated the individual base learner, simple averaging ensemble, and stacking ensemble methods on the publicly available Chapman University and Shaoxing People’s Hospital dataset. The data was split into three parts: 80% for training, 10% for validation, and 10% for testing. As represented in Table 1, the samples of each class are imbalanced. Therefore, we considered a weighted averaging technique instead of a macroscopic averaging technique when evaluating the performance measures such as the area under the ROC curve (AUC), sensitivity, precision, and F1-score. The weighted averaging calculates a measure of performance for each class and then calculates a weighted mean. The weight is determined by the number of samples in each class relative to the total number of samples.

In Table 3, the performances of ResNet-50 based on scalogram images were compared to the performance of ResNet-50 based on ECG grayscale images for each lead. For scalogram images, Lead II demonstrated the highest accuracy (92.24%), AUC (0.991), sensitivity (0.922), precision (0.916), and F1-score (0.916) among the 12 leads. On the other hand, for ECG grayscale images, the aVR lead achieved the highest accuracy (90.90%), sensitivity (0.909), precision (0.911), and F1-score (0.909), while the V1 lead obtained the highest AUC (0.989). Comparing the performance of individual ResNet-50 models for each lead, the model based on scalogram images generally exhibited superior performance.

For the single-modal ensemble methods, single-modal stacking ensemble methods achieved better results than the single-modal simple averaging ensemble and 12 individual base learners for both scalogram images and ECG grayscale images, as described in Table 4. For scalogram images, single-modal stacking ensembles with four machine learning algorithms showed the following diagnostic performance: AUC (ranging from 0.993 to 0.995), accuracy (ranging from 92.34 to 93.01), sensitivity (ranging from 0.923 to 0.930), precision (ranging from 0.915 to 0.925), and F1-score (ranging from 0.913 to 0.925). For ECG grayscale images, single-modal stacking ensembles with four machine learning algorithms achieved the following: AUC (0.993), accuracy (ranging from 92.34 to 93.01), sensitivity (ranging from 0.923 to 0.930), precision (ranging from 0.918 to 0.925), and F1-score (ranging from 0.917 to 0.924). Comparing the scalogram image and the ECG grayscale image, both single-modal stacking ensemble methods showed similar performance. However, random forest and XGBoost showed better results in scalogram images, and logistic regression showed better results in ECG grayscale images.

For the multi-modal stacking ensemble method, the best accuracy (93.97%), sensitivity (0.940), precision (0.937), and F1-score (0.936) were obtained when logistic regression was used as a meta learner as shown in Table 5. In addition, we could obtain the best AUC (0.996) when XGBoost was used as a meta learner. Compared with LSTM, BiLSTM, individual base learners, and single-modal ensemble methods, the proposed multi-modal ensemble methods showed better diagnostic performances. In Figure 4, we represented confusion matrices of two individual leads, a single-modal stacking ensemble with random forest for scalogram images, a single-modal stacking ensemble with logistic regression for ECG grayscale images, multi-modal simple averaging ensemble, and a multi-modal stacking ensemble with logistic regression for comparison.

## 4. Discussion

In this study, we proposed a multi-modal stacking ensemble which combines information from different two modalities, scalogram images and ECG grayscale images. The ResNet-50 model was used as the individual base learner of the stacking ensemble, and one of the machine learning algorithms, logistic regression, SVM, random forest, and XGBoost was utilized as the meta learner. Logistic regression exhibited the highest accuracy, sensitivity, precision, and F1-score and XGBoost achieved the best AUC among the four machine learning algorithms when employed as a meta learner.

The proposed multi-modal stacking ensemble relies on the predictions obtained from both the ECG grayscale image and the scalogram image to generate final predictions. The ECG grayscale image provides cardiologists with information similar to a patient’s ECG graph displayed on a monitor, while the scalogram image offers information about the time-frequency relationship of the ECG signals. In other words, the proposed model has the advantage of collecting multi-modal information potentially contained in the ECG grayscale image and the scalogram image, thereby enabling more accurate predictions of CVDs. From a practical perspective, the utilization of multi-modal information can be crucial for improving the accuracy of predictions in medical environments where accuracy is of utmost importance.

There are many studies that have applied ensemble algorithms to the healthcare field. Kang et al. improved the AUC by simply averaging the predictions from five CNN algorithms (ResNet-101, Xception, Inception-v3, InceptionResNet-v2, DenseNet-201) in classifying breast microcalcification in screening mammograms [31]. Abdar et al. introduced a two-layer nested ensemble method that employed stacking and voting as the classifier to identify benign breast tumors from malignant cancers. Their results indicated that the proposed ensemble algorithms achieved higher performance than single classifiers and most of the previous works [32]. Rao et el. proposed an ensemble model, which integrates three CNNs (DenseNet-121, Inception-v3, and InceptionResnet-v2) in a novel way. The proposed ensemble model showed better performance than the traditional ensemble technique in predicting the recurrence of odontogenic keratocysts (OKCs) on a small chunk of biopsy [33].

There are various public ECG databases on the problem of arrhythmia classification: MIT-BIH arrhythmia database, CinC/Physionet Challenge 2017 database (CinC2017), China Physiological Signal Challenge 2018 database (CPSC2018), PTB-XL database, and Chapman University and Shaoxing People’s Hospital arrhythmia database [6,34,35,36,37]. Among these databases, some of the researchers employed the same database, the Chapman University and Shaoxing People’s Hospital arrhythmia database, that we analyzed. Yildirim et al. constructed an efficient DNN model combining 1D CNN and LSTM and achieved a 92.24% accuracy [11]. Merdjanovska et al. adopted the CPSCWinnerNet model, the winning model of the 2018 China Physiological Signal Challenge, consisting of convolutional blocks, GRUs, and an attention layer. They achieved an accuracy of 94.00% [38]. Baygin et al. proposed a novel classification model which generated 16,384 multilevel features using homeomorphically irreducible tree and maximum absolute pooling. The Chi2 feature selector was used to select the 1000 most informative features, which were subsequently classified using the SVM classifier. The model showed a 92.95% accuracy despite being a feature-based method rather than an end-to-end method [39]. Guan et al. presented a new approach called the hidden attention residual network (HA-ResNet) for the automated classification of arrhythmia. They used three different images, Recurrence Plot, Gramian Angular Field, and Markov Transition Field, as input images which were converted from 1D ECG. The Ha-ResNet algorithm achieved an F1-score of 0.876, a sensitivity of 0.882, and a precision of 0.876 [40]. It is prudent to be careful when comparing directly to the studies mentioned above due to differences in the test data. However, our proposed multi-modal stacking ensemble achieved comparable performance.

Despite demonstrating reasonable performance, this study has some limitations. First, with the exception of LSTM and BiLSTM, the majority of the experiments covered in the study are based on 2D CNNs. We compared the proposed method with base learners and single-modal ensemble methods to show the effectiveness of the proposed multi-modal stacking ensemble. However, it would also be worthwhile to compare the proposed method with feature-based machine learning algorithms or 1D CNN models. The second limitation pertains to the dataset utilized in this study. The 12-lead ECG arrhythmia database collected by Chapman University and Shaoxing People’s Hospital is based on severely imbalanced data. As described in Table 1, the SB category has 3888 samples, while the SI category only contains 397 samples. In order to alleviate this problem, we evaluated the performance measures with a weighted averaging technique instead of a macroscopic averaging technique. To address this issue, one could consider using several large publicly available ECG data sets, such as the recently published PTB-XL [37]. Third, when constructing the stacking ensemble, only one 2D CNN algorithm, ResNet-50, was used as the base learner. It would be necessary to optimize the architecture of the proposed model with a variety of combinations of deep learning and machine learning algorithms.

## 5. Conclusions

In this study, we proposed the use of a multi-modal stacking ensemble for the prediction of CVDs. The proposed method achieved superior performance compared to LSTM, BiLSTM, individual base learner, simple averaging ensemble, and single-modal stacking ensemble methods. These results suggest that a multi-modal stacking ensemble may be a promising approach for improving the accuracy of CVD prediction. Further research is needed to explore the use of multi-modal stacking ensemble methods with large ECG datasets and other combinations of 2D CNNs and machine learning algorithms.

## Figures and Tables

**Figure 1 jpm-13-00373-f001:**
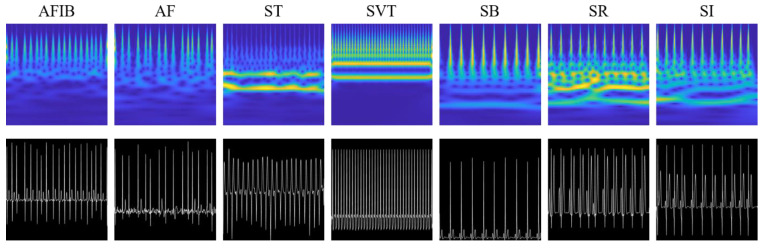
Sample images of scalogram and ECG grayscale images for the 7 groups of ECG recordings (AFIB, AF, ST, SVT, SB, SR, and SI). Each column represents the AFIB, AF, ST, SVT, SB, SR, and SI categories. The first row shows scalogram images and the second row displays grayscale images of the ECG signals.

**Figure 2 jpm-13-00373-f002:**
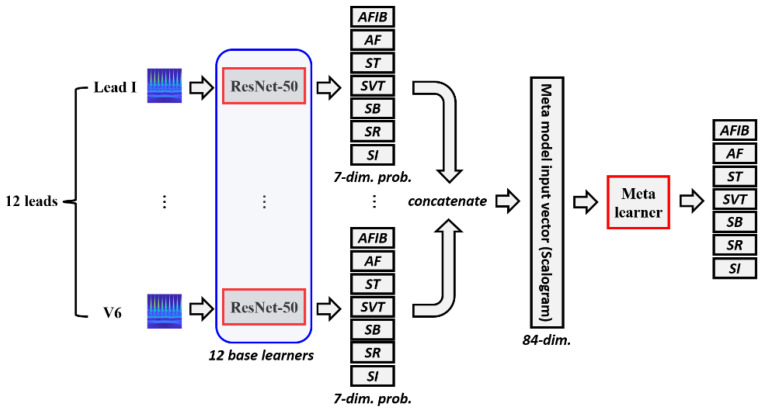
Architecture of single-modal stacking ensemble for scalogram images. For each lead, scalogram images are fed to a pretrained ResNet-50 model. The 7-dimensional output probability vector from each lead is concatenated to make an 84-dimensional vector. Then, the 84-dimensional vector is fed into a meta learner that outputs prediction values for the 7 classes.

**Figure 3 jpm-13-00373-f003:**
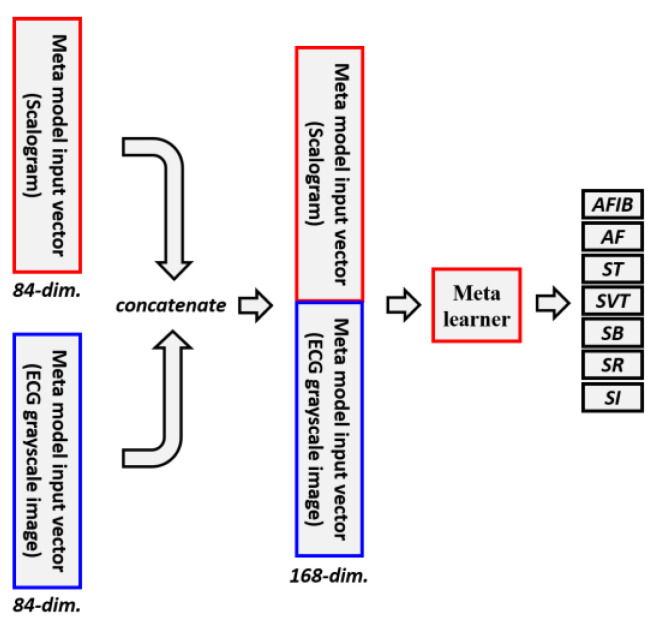
Architecture of multi-modal stacking ensemble. Multi-modal stacking ensemble combines output probabilities of 12 base learners that were trained with scalogram images and ECG grayscale images.

**Figure 4 jpm-13-00373-f004:**
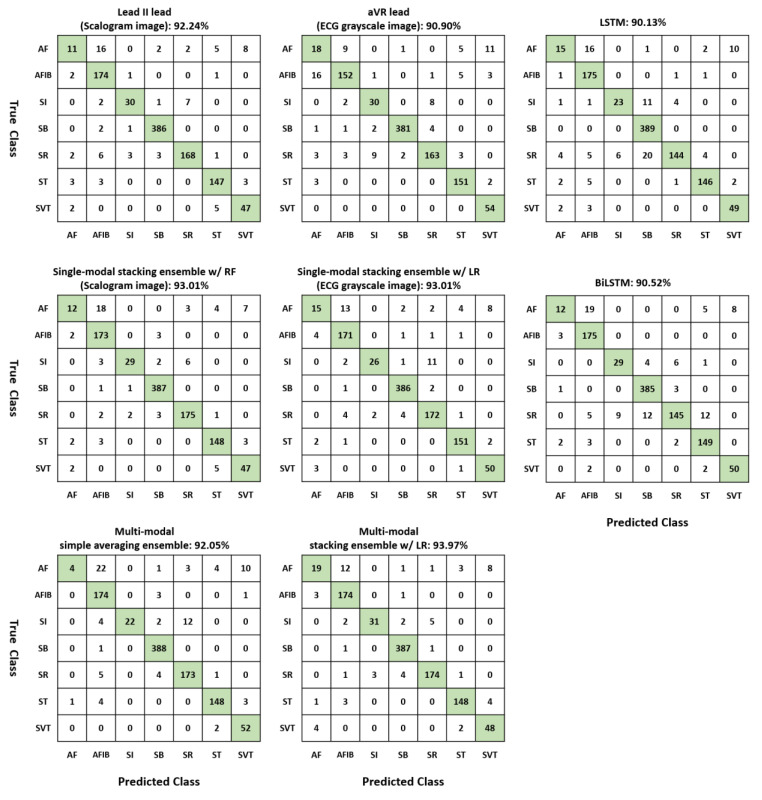
Confusion matrices of two single leads (Lead II lead for scalogram image and aVR lead for ECG grayscale image), LSTM, BiLSTM, a single-modal stacking ensemble with random forest (RF) for scalogram images, a single-modal stacking ensemble with logistic regression (LR) for ECG grayscale images, multi-modal simple averaging ensemble, and a multi-modal stacking ensemble with logistic regression.

**Table 1 jpm-13-00373-t001:** Information on the 7 ECG rhythms.

11 ECG Rhythms	Number of Subjects	Number of Training Data	Number of Validation Data	Number of Test Data
Atrial Fibrillation (AFIB)	1780	1424	178	178
Atrial Flutter (AF)	438	350	44	44
Sinus Tachycardia (ST)	1564	1251	157	156
Supraventricular Tachycardia (SVT)	544	435	55	54
Sinus Bradycardia (SB)	3888	3110	389	389
Sinus Rhythm (SR)	1825	1460	182	183
Sinus Irregularity (SI)	397	318	39	40

**Table 2 jpm-13-00373-t002:** Hyperparameters used in meta learners. LR, logistic regression; SVM, support vector machines; RF, random forest.

Meta Learner Classifier	Hyperparameters in Scikit-Learn	Hyperparameter Ranges
LR	C	1e-3, 1e-2, 1e-1, 1, 1e1, 1e2, 1e3
SVM	C gamma	1e-3, 1e-2, 1e-1, 1, 1e1, 1e2, 1e3 1e-3, 1e-2, 1e-1, 1, 1e1, 1e2, 1e3
RF	n_estimators max_depth max_features	100, 200, 300, 500, 1000, 2000, 3000 5, 10, 15, 20, None ‘log2′, ‘sqrt’
XGBoost	n_estimators max_depth learning_rate	100, 300, 500, 1000 3, 5, 7, 9 0.1, 0.05, 0.01

**Table 3 jpm-13-00373-t003:** Diagnostic performance of individual base learners for scalogram and ECG grayscale images.

	Scalogram Image					ECG Grayscale Image				
Lead Names	AUC	ACC (%)	SEN	PRE	F1-Score	AUC	ACC (%)	SEN	PRE	F1-Score
Lead I	0.985	88.89	0.889	0.871	0.875	0.981	88.41	0.884	0.867	0.872
Lead II	**0.991**	**92.24**	**0.922**	**0.916**	**0.916**	0.988	89.56	0.896	0.888	0.889
Lead III	0.988	90.52	0.905	0.899	0.899	0.987	89.27	0.893	0.886	0.886
aVR	0.986	88.79	0.888	0.889	0.886	0.985	**90.90**	**0.909**	**0.911**	**0.909**
aVL	0.984	88.79	0.888	0.859	0.870	0.981	86.97	0.870	0.868	0.851
aVF	0.990	90.52	0.905	0.904	0.895	0.983	88.60	0.886	0.877	0.873
V1	0.987	88.51	0.885	0.872	0.874	**0.989**	90.71	0.907	0.903	0.900
V2	0.981	89.75	0.898	0.892	0.892	0.979	88.51	0.885	0.867	0.867
V3	0.982	89.66	0.897	0.883	0.885	0.978	87.26	0.873	0.866	0.864
V4	0.981	89.08	0.891	0.872	0.878	0.977	89.46	0.895	0.883	0.887
V5	0.974	88.22	0.882	0.878	0.877	0.976	87.07	0.871	0.859	0.862
V6	0.976	88.03	0.880	0.872	0.870	0.980	87.07	0.871	0.856	0.859

ACC, accuracy; SEN, sensitivity; PRE, precision. Highest values are in bold.

**Table 4 jpm-13-00373-t004:** Diagnostic performance of two single leads (Lead II lead for scalogram image and aVR lead for ECG grayscale image), single-modal simple averaging ensemble, and single-modal stacking ensemble methods for scalogram and ECG grayscale images.

	Scalogram Image					ECG Grayscale Image				
	AUC	ACC (%)	SEN	PRE	F1-Score	AUC	ACC (%)	SEN	PRE	F1-Score
Single lead	0.991	92.24	0.922	0.916	0.916	0.985	90.90	0.909	0.911	0.909
Simple averaging ensemble	0.993	91.95	0.920	0.912	0.907	**0.993**	91.95	0.920	**0.927**	0.904
Stacking ensemble (LR)	0.994	92.72	0.927	0.922	0.921	**0.993**	**93.01**	**0.930**	0.925	**0.924**
Stacking ensemble (SVM)	0.993	92.34	0.923	0.915	0.913	**0.993**	92.53	0.925	0.919	0.917
Stacking ensemble (RF)	0.993	**93.01**	**0.930**	**0.925**	0.923	**0.993**	92.53	0.925	0.918	0.918
Stacking ensemble (XGBoost)	**0.995**	92.91	0.929	0.924	**0.925**	**0.993**	92.34	0.923	0.918	0.919

LR, SVM, RF, and XGBoost indicate the machine learning algorithms used in the meta learner. ACC, accuracy; SEN, sensitivity; PRE, precision; LR, logistic regression; SVM, support vector machines; RF, random forest. Highest values are in bold.

**Table 5 jpm-13-00373-t005:** Diagnostic performance of LSTM, BiLSTM, multi-modal simple averaging ensemble, and multi-modal stacking ensemble methods.

	AUC	ACC (%)	SEN	PRE	F1-Score
LSTM	0.976	90.13	0.901	0.898	0.894
BiLSTM	0.974	90.52	0.905	0.901	0.897
Multi-modal simple averaging ensemble	0.995	92.05	0.920	0.921	0.905
Multi-modal stacking ensemble (LR)	0.995	**93.97**	**0.940**	**0.937**	**0.936**
Multi-modal stacking ensemble (SVM)	0.995	93.39	0.934	0.930	0.928
Multi-modal stacking ensemble (RF)	0.995	93.58	0.936	0.929	0.933
Multi-modal stacking ensemble (XGBoost)	**0.996**	93.68	0.937	0.933	0.933

LR, SVM, RF, and XGBoost indicate the machine learning algorithms used in a meta learner. ACC, accuracy; SEN, sensitivity; PRE, precision; LR, logistic regression; SVM, support vector machines; RF, random forest. Highest values are in bold.

## Data Availability

The dataset is accessible at https://figshare.com/collections/ChapmanECG/4560497/2 (accessed on 19 February 2023).

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
