# Peer review of "Multi-Modal Stacking Ensemble for the Diagnosis of Cardiovascular Diseases"

_jpm, 2023, doi:10.3390/jpm13020373_

Round 1
Reviewer 1 Report
Yoon and Kang presented an ensemble of a ResNet model in combination with different meta learners for the diagnosis of cardiovascular diseases (CVD). The authors give a comprehensive overview of the state of the art of CNNs for diagnosing CVDs. In summary the paper is written in a clear and understandable language and the use of scalogram images in combination with the ECG grayscale image leads to promising results for seven ECG rhythms even on the relatively small dataset of around 10 000 ECG.
Some major points:
· - The grid search could be more extensive, e.g., for the RF only the number of trees were tuned.
· - Can the model generalize? This may be tested e. g. by using another public database.
And please address the following questions:
· - Is the code available?
Some minor points:
· - Page 1, Abstract: “… that that …”
· - Tab. 3 shows the performance on the test set?
· - Page 10 “… 0.913). up to 0.925). For ECG …”
Author Response
Response to Reviewer 1 Comments
The authors would like to thank the Reviewer for their comments. Care has been taken to improve the work and address their concerns as per the specific comments below.
“ Yoon and Kang presented an ensemble of a ResNet model in combination with different meta learners for the diagnosis of cardiovascular diseases (CVD). The authors give a comprehensive overview of the state of the art of CNNs for diagnosing CVDs. In summary the paper is written in a clear and understandable language and the use of scalogram images in combination with the ECG grayscale image leads to promising results for seven ECG rhythms even on the relatively small dataset of around 10000 ECG.”
Point 1(major): The grid search could be more extensive, e.g., for the RF only the number of trees were tuned.
Response 1: The SVM and random forest models underwent an extensive grid search, with an additional search for 'gamma' coefficient in SVM and 'max_depth' and 'max_features' in random forest. The manuscript was updated with the results of a thorough grid search conducted on SVM and random forest.
Point 2(major): Can the model generalize? This may be tested e. g. by using another public database.
Response 2: One of the drawbacks of this study is that we had not tried to test the performance of this algorithm to other datasets. Since the grant for this study is about to end (Feb. 2023), we can no longer apply for IRB for other datasets such as PTB-XL[1]. In the next study, we will apply for IRB for other large public data and perform experiments to generalize the algorithm. We are very sorry for this.
[1] Wagner, P. et al. PTB-XL, a large publicly available electrocardiography dataset. Sci. Data 7, 154 (2020).
Point 3(major): Is the code available?
Response 3: Yes. The code will be available soon on Github, and the manuscript provides the Github URL.
Point 4(minor): Page 1, Abstract: “… that that …”
Response 4: This has now been corrected as suggested. The authors would like to thank the reviewer for this comment.
Point 5(minor): Tab. 3 shows the performance on the test set?
Response 5: Yes. The diagnostic performance in Tables 3 ~ 5 is on the test set.
Point 6(minor): Page 10 “… 0.913). up to 0.925). For ECG …”
Response 6: This has now been corrected as suggested. The authors would like to thank the reviewer for this comment.
Reviewer 2 Report
The paper presents a new method for classifying seven cardiovascular diseases based on a multi-modal stacking ensemble method. Signals from 12 ECG leads were transformed into scalogram images and grayscale images and fed into the ResNet-50 models. The obtained predictions were then used as inputs for the meta-learner. The obtained results are comparable with those presented in the literature. 1.The details of the Continous Wavelet Transformation should be included (wavelet family, mother wavelet, frequency range, time-frequency resolution, etc.). 2. The number of papers in the literature exploiting deep learning for the classification of different diseases (especially cardiovascular diseases) is increasing, and there are a lot of different solutions. Since the obtained results are comparable with those in the literature, the authors should emphasize what are the advantages of the proposed method compared to other methods and what is the practical and/or scientific value of the proposed method.3. Proposed method should be compared to other deep learning methods known in literature and those based on LSTM. Minor 4. Formatting of numbers should be consistent, e.g., Table 1, 1,780, and 1424. 5. Formatting should be improved in general, e.g. Table 2 and Table 3, the title is indented.
Author Response
The authors would like to thank the Reviewer for their comments. Care has been taken to improve the work and address their concerns as per the specific comments below.
“ The paper presents a new method for classifying seven cardiovascular diseases based on a multi-modal stacking ensemble method. Signals from 12 ECG leads were transformed into scalogram images and grayscale images and fed into the ResNet-50 models. The obtained predictions were then used as inputs for the meta-learner. The obtained results are comparable with those presented in the literature.”
Point 1(major): The details of the Continuous Wavelet Transformation should be included (wavelet family, mother wavelet, frequency range, time-frequency resolution, etc.).
Response 1: The information on CWT has now been added in the manuscript as follows:
“An analytic Morse wavelet with a symmetry parameter of 3 (γ = 3) and a time-bandwidth product of 60 ( ) was used to obtain the CWT. The Morse wavelet is perfectly symmetric in the frequency domain and has zero skewness when γ equals 3. The CWT was calculated using 10 voices per octave, a 500Hz sampling frequency, and a signal length of 5,000. The minimum and maximum scales were determined automatically based on the wavelet's energy spread in time and frequency.”
Point 2(major): The number of papers in the literature exploiting deep learning for the classification of different diseases (especially cardiovascular diseases) is increasing, and there are a lot of different solutions. Since the obtained results are comparable with those in the literature, the authors should emphasize what are the advantages of the proposed method compared to other methods and what is the practical and/or scientific value of the proposed method.
Response 2: We have included the following paragraph in the Discussion section:
“The proposed multi-modal stacking ensemble relies on the predictions obtained from both the ECG grayscale image and the scalogram image to generate final predictions. The ECG grayscale image provides cardiologists with information similar to a patient's ECG graph displayed on a monitor, while the scalogram image offers information about the time-frequency relationship of the ECG signals. In other words, the proposed model has the advantage of collecting multi-modal information potentially contained in the ECG grayscale image and the scalogram image, thereby enabling more accurate predictions of CVDs. From a practical perspective, the utilization of multi-modal information can be crucial for improving the accuracy of predictions in medical environments where accuracy is of utmost importance.”
Point 3(major): Proposed method should be compared to other deep learning methods known in literature and those based on LSTM.
Response 3: The proposed method was compared to the LSTM and BiLSTM. The results of LSTM and BiLSTM are in Table 5, and the experimental settings of LSTM and BiLSTM are described in the Materials and Methods section.
Point 4(minor): Formatting of numbers should be consistent, e.g., Table 1, 1,780, and 1424.
Response 4: This has now been corrected as suggested. The authors would like to thank the reviewer for this comment.
Point 5: Formatting should be improved in general, e.g. Table 2 and Table 3, the title is indented.
Response 5: All Table titles were not indented prior to submission of this manuscript. Judging from the fact that the positions of figures and tables have changed after submission, it seems that the indentation of titles is a JPM policy. Based on our prior experience, we are aware that the MDPI proofreading team examines this aspect before the study is published.
Round 2
Reviewer 2 Report
Authors did major revision due to recommendations.
Paper is improved and can be accepted.